Optimal 1-NN prototypes for pathological geometries

http://orcid.org/0000-0003-4121-7479 Sucholutsky Ilia isucholu@uwaterloo.ca
Schonlau Matthias
Department of Statistics and Actuarial Science, University of Waterloo , Waterloo, ON , Canada
Qu Rong
Electronic publication date: 2021 Apr 9
Publication date: 2021
Volume: 7
Electronic Location ID: e464
Received 2020 Nov 5; Accepted 2021 Mar 11
Copyright: © 2021 Sucholutsky and Schonlau
Copyright year: 2021
Copyright holder: Sucholutsky and Schonlau
License: This is an open access article distributed under the terms of the Creative Commons Attribution License, which permits unrestricted use, distribution, reproduction and adaptation in any medium and for any purpose provided that it is properly attributed. For attribution, the original author(s), title, publication source (PeerJ Computer Science) and either DOI or URL of the article must be cited.
License URL: https://creativecommons.org/licenses/by/4.0/

Keywords: Prototype generation, k nearest neighbors, kNN, Prototype selection, Concentric circles

Funding: The authors received no funding for this work.

==============================
Using prototype methods to reduce the size of training datasets can drastically reduce the computational cost of classification with instance-based learning algorithms like the k-Nearest Neighbour classifier. The number and distribution of prototypes required for the classifier to match its original performance is intimately related to the geometry of the training data. As a result, it is often difficult to find the optimal prototypes for a given dataset, and heuristic algorithms are used instead. However, we consider a particularly challenging setting where commonly used heuristic algorithms fail to find suitable prototypes and show that the optimal number of prototypes can instead be found analytically. We also propose an algorithm for finding nearly-optimal prototypes in this setting, and use it to empirically validate the theoretical results. Finally, we show that a parametric prototype generation method that normally cannot solve this pathological setting can actually find optimal prototypes when combined with the results of our theoretical analysis.

Background

The k-Nearest Neighbour (kNN) classifier is a simple but powerful classification algorithm. There are numerous variants and extensions of kNN (Dudani, 1976; Yigit, 2015; Sun, Qiao & Cheng, 2016; Kanjanatarakul, Kuson & Denoeux, 2018; Gweon, Schonlau & Steiner, 2019), but the simplest version is the 1NN classifier which assigns a target point to a class based only on the class of its nearest labeled neighbor. Unfortunately, the family of kNN classifiers can be computationally expensive when working with large datasets, as the nearest neighbors must be located for every point that needs to be classified. This has led to the development of prototype selection methods and generation methods which aim to produce a small set of prototypes that represent the training data (Bezdek & Kuncheva, 2001; Triguero et al., 2011; Bien & Tibshirani, 2011; Garcia et al., 2012; Kusner et al., 2014). Using prototype methods speeds up the kNN classification step considerably as new points can be classified by finding their nearest neighbors among the small number of prototypes. Prototype selection methods select a subset of real points to use as prototypes while prototype generation methods are not similarly restricted and instead create synthetic points (that are not necessarily found in the original data) to act as prototypes. Generating synthetic prototypes allows for more efficient representations so our analysis focuses specifically on the generation of optimal or near-optimal prototypes. The number of prototypes required to represent the training data can be several orders of magnitude smaller than the number of samples in the original training data. Sucholutsky & Schonlau (2020) showed that by assigning label distributions to each prototype, the number of prototypes may even be reduced to be less than the number of classes in the data. This result was demonstrated on a synthetic dataset consisting of N concentric circles where the points on each circle belong to a different class. The authors found that commonly used prototype generation methods failed to find prototypes that would adequately represent this dataset, suggesting that the dataset exhibits pathological geometries. Further analysis revealed that the soft-label kNN variant required only a fixed number of prototypes to separate any number of these circular classes, while the number of prototypes required by 1NN was shown to have an upper bound of about tπ for the tth circle as can be seen in Fig. 1. However, this upper bound did not account for the possibility of rotating prototypes on adjacent circles as a method of reducing the number of required prototypes. We explore this direction to analytically find tighter bounds and an approximate solution for the minimal number of prototypes required for a 1-Nearest Neighbor classifier to perfectly separate each class after being fitted on the prototypes. In particular, we show that this problem actually consists of two sub-problems, or cases, only one of which is closely approximated by the previously proposed upper bound. We also propose an algorithm for finding nearly-optimal prototypes and use it to empirically confirm our theoretical results.

Figure 1 1NN decision boundaries when fitted on ⌈tπ⌉ prototypes per class.

Each shaded circle represents a different class and the outlined points represent the assigned prototypes. The colored regions correspond to the decision boundaries created by the 1NN classifier. The axes form a Cartesian plane whose origin coincides with the smallest class.

Theory

Preliminaries

We first proceed to formalize the problem of having a 1-NN classifier separate the classes after being fitted on a minimal number of prototypes. Consistent with Sucholutsky & Schonlau (2020), we define the tth circle as having radius tc for t = 0,1,…. Because each class is fully separated from non-adjacent classes by its adjacent classes, it is sufficient to consider arbitrary pairs of adjacent classes when trying to find the optimal prototypes. For the rest of this section, we consider arbitrarily selected circles t and t + 1 with the following radii.

r1=tc,r2=(t+1)c,t∈N0,c∈R>0,

Because of the symmetry of each circle, we require that the prototypes assigned to each circle be spaced evenly around it. We assume that circle t and t + 1 are assigned m and n prototypes respectively. We define θ* as the angle by which the prototypes on circle t + 1 are shifted relative to the prototypes on circle t. We record the locations of these prototypes in Cartesian coordinates.

ai=(r1cos⁡(2πim),r1sin⁡(2πim)),i=1,...,ma

bj=(r2cos⁡(2πjn+θ∗),r2sin⁡(2πin+θ∗)),i=1,...,n

We can then find the arc-midpoints of these prototypes as follows.

ai∗=(r1cos⁡(2πi+πm),r1sin⁡(2πi+πm)),i=1,...,m

bj∗=(r2cos⁡(2πj+πn+θ∗),r2sin⁡(2πi+πn+θ∗)),i=1,...,n

Letting d(x,y) be the Euclidean distance between points x and y, we find the distances between prototypes on the same circle.

da(m)=d(ai,ai∗)=2t2c2−2t2c2cos⁡(πm)

db(n)=d(bi,bi∗)=2(t+1)2c2−2(t+1)2c2cos⁡(πn)

We also find the shortest distance between prototypes of circle t and arc-midpoints of circle t + 1 and vice-versa.

d1∗(m,n,θ∗)=mini,j{d(ai,bj∗)∥ i=1,…,mj=1,…,n}=mini,j{t2c2+(t+1)2c2−2t(t+1)c2cos(2πim−2πj+πn−θ∗)∥ i=1,…,mj=1,…,n}

d2∗(m,n,θ∗)=mini,j{d(ai∗,bj)∥ i=1,…,mj=1,…,n}=mini,j{t2c2+(t+1)2c2−2t(t+1)c2cos(2πi+πm−2πjn−θ∗)∥ i=1,…,mj=1,…,n}

The necessary and sufficient condition for the 1-NN classifier to achieve perfect separation is that the distance between prototypes and arc-midpoints assigned to the same circle, be less than the minimal distance between any arc-midpoint of that circle and any prototype of an adjacent circle. This must hold for every circle. Given these conditions and some fixed number of prototypes assigned to the tth circle, we wish to minimize n by optimizing over θ*.

Givenm,tminθ∗⁡n

s.t.d1∗(m,n,θ∗)>db(n)

d2∗(m,n,θ∗)>da(m)

Inspecting the inequalities, we see that they can be reduced to the following system which we note is now independent of the constant c.

(1) −2t+12(t+1)>tcos⁡(2πim−2πj+πn−θ∗)−(t+1)cos⁡(πn)

(2) 2t+12t>(t+1)cos⁡(2πi+πm−2πjn−θ∗)−tcos⁡(πm)

It is clear that n ≥ m, but we separate this system into two cases, n = m and n > m, as the resulting sub-problems will have very different assumptions and solutions. The simpler case is where every circle is assigned the same number of prototypes; however, the total number of circles must be finite and known in advance. In the second case where larger circles are assigned more prototypes, we assume that the number of circles is countable but not known in advance. We also note that for t = 0, a circle with radius 0, exactly one prototype is required. Given this starting point, it can be trivially shown that for t = 1, a minimum of four prototypes are required to satisfy the conditions above (three if the strict inequalities are relaxed to allow equality). However for larger values of t, careful analysis is required to determine the minimal number of required prototypes.

Upper bounds

We first show how our setup can be used to derive the upper bound that was found by Sucholutsky & Schonlau (2020).

Theorem 1 (Previous Upper Bound) The minimum number of prototypes required to perfectly separate N concentric circles is bounded from above by approximately ∑t=1Ntπ, if the number of circles is not known in advance (each circle must have a different number of assigned prototypes).

Proof. Given the setup above, we first consider the worst case scenario where a θ* is selected such that cos⁡(2πim−2πj+πn−θ∗)=cos⁡(2πi+πm−2πjn−θ∗)=cos(0)=1. We can then solve Inequality 1 for n and Inequality 2 for m.

−2t+12(t+1)>tcos⁡(0)−(t+1)cos⁡(πn)

cos⁡(πn)>2(t+1)2−12(t+1)2

n>πarccos⁡(2(t+1)2−12(t+1)2)≈(t+1)π

2t+12t>(t+1)cos⁡(0)−tcos⁡(πm)

cos⁡(πm)>2t2−12t2

m>πarccos⁡(2t2−12t2)≈tπ

This is exactly the previously discovered upper bound.

However, note that we assumed that there exists such a θ*, but this may not always be the case for n>m. If we instead use the same number of prototypes for each circle (i.e., m = n), then we can always set θ∗=πn. This results in a configuration where every circle is assigned n=πarccos⁡(2(t+1)2−12(t+1)2)≈(t+1)π prototypes. While the minimum number of prototypes required on the tth circle remains the same, the total minimum number of prototypes required to separate N circles is higher as each smaller circle is assigned the same number of prototypes as the largest one.

Corollary 2 (Upper Bound—Same Number of Prototypes on Each Circle) The minimum number of prototypes required to perfectly separate N concentric circles is bounded from above by approximately N2π, if the number of circles is finite and known in advance (each circle can have the same number of assigned prototypes).

Lower bounds

An advantage of our formulation of the problem is that it also enables us to search for lower bounds by modifying the θ* parameter. We can investigate the scenario where a θ* is selected that simultaneously maximizes d*1(m,n,θ*) and d*1(m,n,θ*).

Theorem 3 (Lower Bound) The minimum number of prototypes required to perfectly separate N concentric circles is bounded from below by approximately ∑t=1Nt12π, if the number of circles is not known in advance (each circle must have a different number of assigned prototypes).

Proof. If m ≠ n, the best case would be a θ* such that cos⁡(2πim−2πj+πn−θ∗)=cos⁡(2πi+πm−2πjn−θ∗)=cos(πn). Solving the inequalities leads to the following values for m and n.

n>πarccos⁡(2t+12(t+1))≈(t+1)12π

m>πarccos⁡(2t2−t−12t2)≈t(t+1)12π

We note again that such a θ* may not always exist.

Exact and approximate solutions

In the case where m = n, we can always choose a θ* such that cos⁡(2πim−2πj+πn−θ∗)=cos(πn). Solving the inequalities, we get that n>πarccos⁡(2t+12(t+1))≈(t+1)12π. Thus we have a tight bound for this case.

Corollary 4 (Exact Solution—Same Number of Prototypes on Each Circle) The minimum number of prototypes required to perfectly separate N concentric circles is approximately N32π, if the number of circles is finite and known in advance (each circle can have the same number of assigned prototypes).

When m > n, we have that cos⁡(2πim−2πj+πn−θ∗)>cos(πn) as 2πim−2πjn=2πc1gcd(m,n)mn,c1∈N0. Let q:=2πgcd(m,n)mn, then |2πim−2πj+πn−θ∗|≤q2 and |2πi+πm−2πjn−θ∗|≤q2. Thus cos⁡(2πim−2πj+πn−θ∗)≥cos(q2),andcos⁡(2πi+πm−2πjn−θ∗)≥cos(q2).

Using the series expansion at q = 0 we can find that cos⁡(q2)=1−q28+q4384−q646080+O(q8).

Theorem 5 (First Order Approximation—Different Number of Prototypes on Each Circle) The minimum number of prototypes required to perfectly separate N concentric circles is approximately 1+∑t=1Ntπ, if the number of circles is not known in advance (each circle must have a different number of assigned prototypes).

Proof. For a first order approximation, we consider cos⁡(q2)=1−q28+O(q4) and cos⁡(πn)=1−π22n2+O(1n4). Inequality 1 then becomes the following.

−2t+12(t+1)>t(1−q28+O(q4))−(t+1)(1−π22n2+O(1n4))=−1−π22n2(tgcd(m,n)2m2−t−1)+O(1n4)

n2>−π2(t+1)(tgcd(m,n)2m2−t−1)+O(1n2)

However, we know from our previous upper bound that m + 1 ≤ n ≤ m + 4.

Thus 4(n−4)2>gcd(m,n)2m2>1(n−1)2 which means that gcd(m,n)2m2=O(1n2).

n2>−π2(t+1)(tgcd(m,n)2m2−t−1)+O(1n2)=π2(t+1)2+O(1n2)

Therefore we have that n+O(1n)>(t+1)π as desired.

We plot the second order approximation alongside the first order approximation from Theorem 5 in Fig. 2, without rounding to show that the two quickly converge even at small values of t. Thus we can be confident that approximately tπ prototypes are required for the tth circle since this approximation quickly approaches the true minimal number of required prototypes as t increases. Since we can only assign a positive integer number of prototypes to each circle, we assign tπ prototypes to the tth circle; this is also shown in Fig. 2. Applying this to the initial condition that the 0th circle is assigned exactly one prototype results in the following sequence of the minimal number of prototypes that must be assigned to each circle. We note that the sequence generated by the second order approximation would be almost identical, but with a 3 replacing the 4.

1,4,7,10,13,16,19,22,26,29,32,35,38,41…

Corollary 6 (Approximate Solution—Different Number of Prototypes on Each Circle) The minimum number of prototypes required to perfectly separate N concentric circles is approximately ∑t=1Ntπ≈N+N(N+1)π2, if the number of circles is not known in advance (each circle must have a different number of assigned prototypes).

Figure 2 First order (before and after discretizing by rounding to nearest integer) and second order approximations for the minimal number of prototypes that must be assigned to circle t.

The approximations are applied to continuous values of t to show the convergence behavior.

Computational results

Algorithm

While Theorem 5 gives us the number of prototypes required for each circle, it does not give us the exact locations of these prototypes. Finding the locations would require us to know θ, the optimal rotation of circle n + 1 relative to circle n. Unfortunately, the equations involving θ depend on greatest common denominator terms. Since this makes it difficult to find explicit analytical solutions, we instead turn to computational methods to find near-optimal prototypes. The theoretical results above enable us to develop computational methods to empirically find the minimum number of required prototypes. Based on the equations derived in the previous section, we propose an iterative, non-parametric algorithm, Algorithm 1, that proceeds from the innermost circle to the outermost one finding a near-optimal number of required prototypes, and their positions, in a greedy manner.

Algorithm 1 FindPUGS algorithm: finding (nearly-optimal) prototypes using greedy search.

Result: Two ordered lists, N and R, of the minimum number of prototypes required for each circle and their rotations relative to the previous circle.	
T ← the number of circles	
c ← the length by which radii should grow	
Algorithm FindPUGS (T, c)	
 N ← [1]	
 R ← [0]	
 for t = 1,2,…,T – 1 do	
  m ← N[−1];	
  n ← m + 1;	
  p ← 0;	
  while True do	
    da←2t2c2−2t2c2cos⁡(πm);	
    db←2(t+1)2c2−2(t+1)2c2cos⁡(πn);	
    for i = 0,1,…,4mn do	
      θ←iπm∗n∗16	
      if d1(t,c,m,n,θ) > db and d2(t,c,m,n,θ) > da then	
        p←n	
        break	
      end	
    end	
    if p>0 then	
      N.append(p);	
      R.append(θ);	
      break;	
    end	
    n←n+1	
  end	
 end	
 return N, R;	
Procedure d1 (t,c,m,n,θ)	
 dists ← [];	
 for i = 0,…,m – 1 do	
   for j = 0,…,n − 1 do	
     dist ←t2c2+(t+1)2c2−2t(t+1)c2cos⁡(2iπm−2jπn−πn−θ);	
     dists.append(dist);	
   end	
 end	
 return min(dists);	
Procedure d2 (t,c,m,n,θ)	
 dists ← [];	
 for i = 0,…,m − 1 do	
   for j = 0,…,n − 1 do	
    dist ←t2c2+(t+1)2c2−2t(t+1)c2cos⁡(2iπm−2jπn+πm−θ);	
    dists.append(dist);	
   end	
 end	
 return min(dists);	

The core of the algorithm consists of three loops: outer, middle, inner. The outer loop iterates over each circle, from smallest to largest. For each circle, m is set to be the minimum number of prototypes found during the loop for the previous circle. The middle loop then iterates over candidate values of n, starting from m + 1 and increasing by one each time, until it reaches the first value of n for which Eqs. (1) and (2) can be simultaneously solved given the current value of m. Whether the equations can be solved is determined in the inner loop which plugs different values of the rotation angle θ into the system. Since the distance equations are periodic over values of θ, and the length of the period depends on the greatest common divisor of m and n, we can speed the search up by only considering an interval of the length of the maximum period and iterating θ by an angle inversely proportional to the product of m and n. To avoid any potential floating-point precision errors, we use a wider than necessary interval and smaller than necessary update size for θ. At the end of each iteration of the outer loop, the n and θ that were found are recorded. We note that the rotation angle θ is relative to the rotation of the previous circle. In other words, the absolute rotation for a given circle can be found by adding its relative rotation to the relative rotations of all the preceding circles.

Our code for this algorithm can be found at the publicly available GitHub repository associated with this paper. As shown above, the choice of c > 0, the constant length by which the radius of each consecutive circle increases, does not affect the number of required prototypes. Nonetheless, we still include c as a parameter in our algorithm to verify correctness. Running the algorithm for some large T, with any choice of c, results in the following sequence.

1,3,6,12,13,16,19,22,26,29,32,35,38,41…

This sequence appears to converge very quickly to the one predicted by our theorem. Curiously, the small differences between the first few steps of the two sequences cancel out and the cumulative number of required prototypes is identical when there are four or more circles. While requiring the algorithm to find numerical solutions to these equations is perhaps not computationally efficient, it does guarantee near-optimal performance, with the only sub-optimal portion occurring at the start of the sequence where the algorithm outputs 1,3,6,12,13 rather than the optimal 1,3,7,10,13 due to its greedy nature.

We visualize two sub-optimal prototype arrangements, and the near-optimal arrangement found by our algorithm, in Fig. 3. The patterns seen in these visualizations are largely dependent on the greatest common divisors of the number of prototypes on adjacent circles, as well as the relative rotations of the prototypes on adjacent circles. The particularly symmetrical patterns in Fig. 3 are a result of the outer three circles having 3, 6, and 12 prototypes respectively, doubling each time. We show another example of the decision boundaries exhibited by 1NN when fitted on near-optimal prototypes in Fig. 4.

Figure 3 1NN decision boundaries when fitted on two sub-optimal prototype arrangements as well as near-optimal prototypes found using the FindPUGS algorithm.

Each shaded circle represents a different class and the outlined points represent the assigned prototypes. The colored regions correspond to the decision boundaries created by the 1NN classifier. The axes form a Cartesian plane whose origin coincides with the smallest class. (A and B) Prototypes on adjacent circles are not optimally rotated resulting in imperfect class separation in certain regions. (C) Prototypes are optimally rotated resulting in perfect class separation.

Figure 4 The ClusterCentroids prototype generation method finds similar prototypes to our proposed algorithm when parametrized with the near-optimal number of prototypes per class.

Heuristic prototype methods

We compare the performance of our proposed algorithm against a variety of existing prototype selection and generation methods. Specifically, we compare against every under-sampling method implemented by Lemaître, Nogueira & Aridas (2017) in the “imbalanced-learn” Python package. We describe the prototype methods below and summarize their key properties in Table 1.

TomekLinks: Rebalances classes by removing any Tomek links (Tomek, 1976b).

RandomUndersampler: Rebalances classes by randomly selecting prototypes from each class.

OneSidedSelection: Rebalances classes by isolating each class and resampling the negative examples (composed of the remaining classes) (Kubat, 1997).

NeighbourhoodCleaningRule: Improves on OneSidedSelection in settings where particularly small classes are present. As a result, it focuses more on improving data quality than reducing the size of the dataset (Laurikkala, 2001).

NearMiss: All three versions rebalance classes by resampling the negative examples for a particular class. V1 selects the points from other classes which have the shortest distance to the nearest three points from the target class. V2 selects the points from other classes which have the shortest distance to the furthest three points from the target class. For every point in the target class, V3 selects a fixed number of the nearest points from other classes (Mani & Zhang, 2003).

InstanceHardnessThreshold: Rebalances classes by fitting a classifier to the data and removing points to which the classifier assigns lower probabilities (Smith, Martinez & Giraud-Carrier, 2014).

EditedNearestNeighbours: Resamples classes by removing points found near class boundaries defined by a fitted classifier (Wilson, 1972).

RepeatedEditedNearestNeighbours: Resamples classes by repeatedly applying EditedNearestNeighbours and refitting the classifier (Tomek, 1976a).

AllKNN: Resamples classes similarly to RepeatedEditedNearestNeighbours but increases the parameter k of the classifier each time (Tomek, 1976a).

CondensedNearestNeighbours: Rebalances classes by repeatedly fitting a 1NN on the set of candidate prototypes and then adding all misclassified points to that set (Hart, 1968).

ClusterCentroids: Rebalances classes by using kMeans to replace clusters with their centroids.

Table 1 A list of prototype selection and generation methods.

The last column describes how the number of prototypes is chosen for each class.

Name	Type	Choosing number of prototypes	
TomekLinks	Selection	Automatic	
RandomUndersampler	Selection	Automatic or Manual	
OneSidedSelection	Selection	Automatic	
NeighbourhoodCleaningRule	Selection	Automatic	
NearMissV1-3	Selection	Automatic or Manual	
InstanceHardnessThreshold	Selection	Automatic or Semi-Automatic	
AllKNN	Selection	Automatic	
EditedNearestNeighbours	Selection	Automatic	
RepeatedEditedNearestNeighbours	Selection	Automatic	
CondensedNearestNeighbours	Selection	Automatic	
ClusterCentroids	Generation	Automatic or Manual	

For each experiment, the dataset consists of 800 points divided as evenly as possible between the circles. We note that most methods are not able to reduce the number of prototypes much lower than the number of training points. This is in part due to the automatic class re-balancing that some of these methods attempt to do. Since all classes already have roughly the same number of points, and since none are misclassified when all 800 training points are used, several of the methods determine that little-to-no re-sampling is necessary. As a result, these methods provide at most a small reduction in the number of prototypes. We visualize some of the methods performing automatic undersampling in Fig. 5, where two common failure modes can be seen: the methods either fail to reduce the number of prototypes but achieve good separation of classes, or reduce the number of prototypes but fail to separate the classes. However, the user can also override the automatic re-balancing for a few of the methods, those which include the “Manual” option in Table 1, by passing the number of desired prototypes per class as a hyperparameter. We pass the optimal number of prototypes per class suggested by our earlier theoretical analysis, and the near-optimal number suggested by our algorithm, to these methods and document the results in Fig. 6. Curiously, none of the prototype selection methods achieve perfect separation when restricted to this nearly optimal number of prototypes, even though the nearly-optimal prototypes found by our algorithm have extremely close-by neighbors among the training points. In other words, it is not theoretically impossible for the prototype selection methods to select prototypes close to where the optimal prototypes would be, and yet they do not. Meanwhile, the ClusterCentroids prototype generation method finds similar prototypes to the ones proposed by our algorithm as seen in Fig. 4.

Figure 5 Examples of failure modes on four and six-class concentric circles data using prototype methods where number of prototypes per class was found automatically (semi-automatically for the InstanceHardnessThreshold method).

Figure 6 Examples of failure modes on four and six-class concentric circles data using prototype methods for which the number of prototypes per class was set manually.

Additional experiments

By using the results of our theoretical analysis to parametrize ClusterCentroids, we enable it to find efficient sets of high-quality prototypes. We combine our proposed algorithm with ClusterCentroids to produce a method that combines the benefits of both: a non-parametric algorithm that finds near-optimal prototypes in our pathological case but is robust to noise. We conduct additional experiments to show that this resulting method is indeed robust to noise. Each experiment still uses a dataset of 800 points that are spread over N concentric, circular classes with radius growth parameter c = 0.5; however, we now introduce Gaussian noise to each class. The level of noise is controlled by parameter σ, the standard deviation of the Gaussian distribution underlying the positioning of points within a class. The ratio of σ to c dictates how much overlap occurs between classes. For example, when σ=0.25=c2, only around 68% of points belonging to a particular class will be contained within the band of thickness c = 0.5 associated with that class. We use four levels of noise (σ = 0.05, 0.1, 0.2, 0.4) and five different numbers of classes (4, 6, 8, 10, 12), for a total of 20 generated datasets to which we apply the near-optimally parametrized ClusterCentroids algorithm and measure classification accuracy.

The results are detailed in Table 2 and visualized in Fig. 7. As expected, increasing noise causes a decrease in classification accuracy. However, the decrease in accuracy is roughly equal to the percentage of points found outside of their class’s band as they are indistinguishable from the points of the class whose band they are in. This suggests that the 1NN classifier fitted on prototypes designed by the near-optimally parametrized ClusterCentroids algorithm, approaches the Bayes error rate. We also note that as the number of classes increases, and hence the number of points per class decreases, the accuracy of the classifier stays stable or even increases at high levels of noise. The near-optimally parametrized ClusterCentroids algorithm is clearly robust to increases in the number of classes. It is also partially robust to noise, even though noise violates the underlying assumptions on which the nearly-optimal parametrization is based.

Table 2 Accuracy of ClusterCentroids parametrized with near-optimal number of prototypes.

Noise (σ)	Out-of-band points (%)	Accuracy (4 classes)	Accuracy (6 classes)	Accuracy (8 classes)	Accuracy (10 classes)	Accuracy (12 classes)	
0.05	0	0.975	0.98625	0.9775	0.9775	0.975	
0.1	1	0.94375	0.94	0.94	0.9425	0.93125	
0.2	21	0.77375	0.77625	0.76375	0.79375	0.81375	
0.4	53	0.53	0.4925	0.5325	0.58375	0.60875	

Figure 7 ClusterCentroids parametrized with near-optimal number of prototypes applied to various levels of noise.

From top to bottom, the rows correspond to 4, 6, 8, 10, and 12 classes. From left to right, columns correspond to s = 0.05, 0.1, 0.2, 0.4.

Conclusion

The kNN classifier is a powerful classification algorithm, but can be computationally expensive. While numerous prototype methods have been proposed to alleviate this problem, their performance is often strongly determined by the underlying geometry of the data. Certain pathological geometries can result in especially poor performance of these heuristic algorithms. We analyzed one such extreme setting and demonstrated that analytical methods can be used to find the minimal number of optimal prototypes required for fitting a 1NN classifier. We also found that in such pathological cases, theoretical analysis may not be able to provide the exact locations of the prototypes, but it can be used to derive systems of equations that when solved with numerical methods, produce optimal or near-optimal prototypes.

To demonstrate this approach, we proposed an algorithm for finding nearly-optimal prototypes in the particular pathological setting of concentric circular classes, and used it to validate our theoretical results. The algorithm outperformed all prototype selection methods that it was tested against. A prototype generation method was able to find the optimal prototypes, but only when parametrized using either the theoretical results or the outputs of our proposed algorithm. We further showed that this combination of our proposed algorithm with an existing prototype generation method exhibited the desirable features of both: it is non-parametric and is guaranteed to find near-optimal prototypes even in the examined pathological case, but it is general enough that it is robust to violations of the underlying assumptions of our theoretical analysis, such as the addition of Gaussian noise to the data.

We believe that identifying and studying further pathological geometries in kNN and other machine learning models is an important direction for understanding their failure modes and jointly improving training algorithms and prototype methods.

Additional Information and Declarations

Competing Interests

Author Contributions

Data Availability

The authors declare that they have no competing interests.

Ilia Sucholutsky conceived and designed the experiments, performed the experiments, analyzed the data, performed the computation work, prepared figures and/or tables, authored or reviewed drafts of the paper, and approved the final draft.

Matthias Schonlau conceived and designed the experiments, authored or reviewed drafts of the paper, supervised and advised, and approved the final draft.

The following information was supplied regarding data availability:

The code is available at GitHub: https://github.com/ilia10000/LO-Shot, in “Code - Paper2 directory”.

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
