# Peer review of "Optimal 1-NN prototypes for pathological geometries"

_PeerJ Computer Science, doi:10.7717/peerj-cs.464_

## Round 0.1 · original submission · Major Revisions

In particular, please carefully address the detailed comments from reviewer 3 to clarify the experimental results thus to support the claims made in the paper.

Reviewer 1 ·

Basic reporting

The paper is well-written and self-contained. It is easy to read. I didn’t find any obvious grammar or spelling mistakes. All figures are clear and easy to understand. Detailed proofs about the upper-bound and lower-bound have been given.

Experimental design

The primary research is within the aims and scope of the journal.
The research question is well defined and interesting.
The paper is theoretically and technically solid.
The algorithm has been clearly described.

Validity of the findings

The theory and method is novel and interesting.
The code is provided.
Conclusions are well stated.
The related works are sufficiently discussed.

Additional comments

The authors investigated an interesting problem that how to find the optimal 1-NN prototypes for pathological geometries. An analytic method and a near-optimal algorithm are proposed. Theoretical analysis about the upper-bound and lower-bound is given. The results are illustrated in figures.


I didn’t see any major problems. But, I encourage the authors to address the following problems:
1. As shown in Figure 3, the optimal solution (data points and decision boundaries) has an interesting structure. Could you give an intuitive explanation why the (colored) areas determinate by decision boundaries have such shape? Why they are not symmetric like the left and middle sub-figures? Can we adjust some hyper-parameters and get results with different visualization appearances?
2. Could you introduce some classic solutions for this problem and discuss why their heuristics failed?
3. It would be better to provide more and diverse results to prove the effectiveness of the proposed algorithm.


Minor problems:
1. The equations between Line 61 and 62 exceeded the page length.
2. Figure 2 shows a linear relationship between the circle t and the number of prototypes for 1st order and 2nd order settings. I think the line should be discrete. Right?

Reviewer 2 ·

Basic reporting

Some notations are not clearly defined, for example the horizontal/vertical axes of Figure 1, Figure 3

Experimental design

no comment

Validity of the findings

The computational results are not sufficient. The optimal prototypes for a given dataset, as well as the comparison between different heuristic algorithms, are not reported.

Additional comments

The implications of the tighter bounds are not very clear. For example how these tighter bounds can be used to find the prototypes more efficiently should be reported. The comparison between the proposed algorithms and different heuristic methods for finding nearly-optimal prototypes should be reported as well.

Reviewer 3 ·

Basic reporting

The paper addresses an important and interesting problem in machine learning and classification, specifically finding so-called ‘prototypes’ which are representatives of data sets, that aim to capture similar (or even better) information and can be used to speed up the computations in k-nearest neighbor. Generally the paper is well-written, self-contained, professional, and clear about what it is saying.

Experimental design

I find that it is somewhat lacking in terms of experimental design in a fashion that is directly relevant to prototypes in machine learning in a way that it claims. The very specific problem of concentric circles was already introduced in a previous paper by the same authors, and the main contribution is to do some exact computations for this one very specific and artificial example. It seems to assume that all data would sit exactly on one of the concentric circles and with diameters precisely multiples, given by the parameter ‘t’ of a fixed size. This seems very unrealistic; It is more likely there would be concentric rings, with different widths, and probably some distortions. It is not at all clear what lessons might be drawn from the exact solutions, and be likely to have any relevance to real cases. Although the general idea of concentric circles is something that could reasonably appear it seems very unlikely that it would be in such a regular structure. Hence, the computations are not matching the experimental design that would be needed. Also, if it were so regular than a reasonable learning system - maybe with the aid of kernels - would be likely to pick this up.
Also, my understanding is that the idea of prototypes is that they select a smaller set from a finite sample of values. Whereas in this case the selection seems to be from all of the points on each circle - essentially an infinite size initial sample. I would have expected to see methods to select from a finite (randomised) selection of points on each circle - then the ‘exact’ solutions given could not be applied directly.

It also does not match the claim in the abstract about it heuristic algorithms failing, as there is no inclusion of experiements to show this.

Hence this example seems to be quite unrealistic, and not very informative, nor relevant to prototypes. It is possible that the authors have some other motivation or context in mind, but then this should be made clear.

Validity of the findings

The limited findings are valid as far as I can tell.

The abstract claims that heuristic algorithms fail in this case, but gives no data or discussion to support this.

Additional comments

minor point:

Theorem 5: should “each circle must have a different number” be “may have”?
Seems it should be relaxing an equality, not adding an alldifferent constraint.

---

## Round 0.2 · accepted · Accept

All the comments from the reviewers have now been addressed satisfactorily. I'd like to recommend accepting the revised manuscript.

Reviewer 3 ·

Basic reporting

pass

Experimental design

pass

Validity of the findings

pass

Additional comments

The changes have greatly improved the paper, and it now makes a lot more sense to me. It does seem reasonable that the high-quality answers for these special and artificial cases can help provide useful challenging benchmarks for more generally-applicable usually-heuristic methods.

Minor issues:

page 4 say ‘countable’ but this makes it sound like theory results on countability, and is presumably not meant? Might simply mean ‘finite’ ?